# Modification and Evaluation of Attention-Based Deep Neural Network for Structural Crack Detection

**DOI:** 10.3390/s23146295

**Published:** 2023-07-11

**Authors:** Hangming Yuan, Tao Jin, Xiaowei Ye

**Affiliations:** 1Polytechnic Institute, Zhejiang University, Hangzhou 310058, China; 2Department of Civil Engineering, Zhejiang University, Hangzhou 310058, China

**Keywords:** structural crack, deep learning, attention mechanism, structural health monitoring

## Abstract

Cracks are one of the safety-evaluation indicators for structures, providing a maintenance basis for the health and safety of structures in service. Most structural inspections rely on visual observation, while bridges rely on traditional methods such as bridge inspection vehicles, which are inefficient and pose safety risks. To alleviate the problem of low efficiency and the high cost of structural health monitoring, deep learning, as a new technology, is increasingly being applied to crack detection and recognition. Focusing on this, the current paper proposes an improved model based on the attention mechanism and the U-Net network for crack-identification research. First, the training results of the two original models, U-Net and lrassp, were compared in the experiment. The results showed that U-Net performed better than lrassp according to various indicators. Therefore, we improved the U-Net network with the attention mechanism. After experimenting with the improved network, we found that the proposed ECA-UNet network increased the Intersection over Union (IOU) and recall indicators compared to the original U-Net network by 0.016 and 0.131, respectively. In practical large-scale structural crack recognition, the proposed model had better recognition performance than the other two models, with almost no errors in identifying noise under the premise of accurately identifying cracks, demonstrating a stronger capacity for crack recognition.

## 1. Introduction

With the development of China’s economy and the continuously expanding investment in infrastructure, the number of large structures such as bridges and buildings has increased [1,2,3]. Some buildings are in a long-term state of overload, corrosion, etc., and are susceptible to functional barriers under the overlapping impact of natural disasters, resulting in serious accidents [4,5]. Cracks in the structure are some of the most important indicators of structural damage or destruction caused by aging and other reasons [6,7]. As time goes by, the width and number of cracks will gradually increase, affecting the safety, practicality, and durability of the structure [8]. If reliable inspections are conducted on cracks, this can effectively prevent serious damage to buildings and prolong the life of facilities through appropriate maintenance [9,10,11,12,13]. Traditional inspection methods mainly rely on visual inspection and bridge inspection vehicles. Among them, the efficiency and accuracy of manual visual inspection [14,15,16] are greatly affected by the experience of the inspectors, and the human eye has many limitations, which can easily cause omissions; bridge inspection vehicles have many safety hazards during operation, and they are prone to high costs, slow efficiency, and traffic congestion. Therefore, to speed up the inspection process and achieve reliable and consistent inspections, deep-learning networks have developed rapidly in the field of structural health, and the crack recognition ability for complex situations continues to improve [17,18].

Deep learning [19,20] has greatly improved the latest technological level in fields such as visual object recognition and object detection, providing precise analysis for crack detection in structures. Semantic segmentation networks such as DeepLab [21], SegNet [22], and FCN [23] have also been widely used in crack recognition and detection. To overcome the limitations of human resources for visual inspections and provide accurate detection of multiple types of crack damage, Cha et al. [24] introduced a detection method based on faster convolutional neural networks. Researchers developed a database containing 2366 images and used it for modification, training, validation, and testing to develop multiple types of damage detection. Due to its fast speed and high accuracy, a video-based near real-time damage-detection framework based on trained networks was proposed. Li et al. [25] established a database of 2750 images of concrete structure cracks, spalling, weathering, etc., which was manually annotated. They tested and compared the fully convolutional network (FCN) architecture using this database and used the SegNet-based method to demonstrate that this method can accurately detect multiple concrete damage areas at the pixel level. Cardellicchio et al. [26] collected an existing image database of defects in reinforced concrete bridges, and domain experts classified the most common types of defects. Several convolutional neural network (CNN) algorithms were applied to the dataset for automatic identification of all defects. Zhang et al. [27] developed a context-aware deep convolutional semantic-segmentation method, leveraging local cross-state and cross-space constraints for image block fusion. Yamane et al. [28] proposed a deep learning-based semantic segmentation method that accurately detects concrete crack regions and removes other artifacts in photographs of concrete structures under adverse conditions. Lee et al. [29] proposed a crack-detection network and crack image generation algorithm based on image-segmentation networks. The training and validation results demonstrate that this method possesses high robustness and accuracy. Li et al. [30] proposed a semi-supervised method for road-crack detection that uses unlabeled road images for training and employs adversarial learning and fully convolutional discriminators to improve accuracy.

U-Net [31], as the most classic representative network of the U-shaped network structure, can extract the input image features. In addition, U-Net’s accuracy is often higher than that of other models, and its structure is simple, mainly divided into three parts: feature extraction, clipping, and upsampling. It is widely used in industrial defect detection and has achieved good results in image segmentation [32,33]. Although U-Net has achieved high segmentation accuracy and speed, traditional convolutional and pooling layers generally suffer from information loss during information transmission, which is affected by the background environment, resulting in blurred boundaries of the segmented target area and a lot of noise. Based on the above shortcomings, we are focusing on increasing the attention of the network on small target features, specifically for crack-detection problems. We propose to embed an attention mechanism into the existing model to improve the ability to recognize cracks [34,35].

Attention mechanisms in deep learning [36,37] are very similar to human visual attention mechanisms, which select more important information for the current target and remove redundant information. This allows the network to adaptively focus on the necessary information and can be achieved by using importance weight vectors to approximate the final target value through weighted vector summation. Attention mechanisms mainly include the SE (Squeeze-and-Excitation) attention mechanism [38,39], the CBAM (Convolutional Block Attention Module) attention mechanism [40], the CA (Channel Attention) attention mechanism [41], etc. The introduction of attention mechanisms can improve crack-image detection accuracy with a small increase in computational cost. This effectively extracts multi-scale features of cracks while capturing local features and the edge details of small cracks. Attention mechanisms can focus on key areas and reconstruct semantics, significantly improving the crack-segmentation ability of the U-Net model [42,43,44,45].

In this article, research on the improvement of model-recognition performance through the addition of attention mechanisms was conducted. A modification has been made to the U-Net using an ECA (Efficient Channel Attention) mechanism, and a performance comparison has been conducted with the original network. The improvements were supported by the indicators and image testing. Large image-recognition experiments were conducted on actual structural cracks, and the recognition results were compared and evaluated.

## 2. Method of Attention-Based Structural Crack Detection

The attention mechanism is similar to our eyes as we use them to focus on the data we want to pay attention to. Similarly, the attention mechanism acts like the eyes of a deep-learning network, which can inform the network about the specific image features that we want to focus on and thus, enable more accurate acquisition of image information. This article focuses on the scientific problem of extracting the semantic segmentation of structural cracks. It mainly compares different deep-learning network models and improves recognition performance by adding attention mechanisms. The ECA attention mechanism proposed by Wang et al. [46] can achieve significant accuracy with a small number of parameters. This module is an efficient attention-channeling module, which can avoid feature loss caused by dimensionality reduction in other attention mechanisms and efficiently capture information interaction between different channels. In terms of its structural characteristics, the ECA attention mechanism is more suitable for network models with simpler structures such as U-Net due to its lightweight structure. The structure of ECA is shown in Figure 1:

The feature map is transformed from a matrix to a 1 × 1 × C vector through average pooling. The formula for the adaptive one-dimensional convolution kernel size k is shown in Formula (1). By adjusting the kernel size, the weight of each channel in the feature map is obtained. Then, the obtained weights are multiplied with each channel of the original input image to obtain the feature map with attention added.
k = |log_2_(C)/γ + b/γ|; γ = 2, b = 1(1)

Based on the research content of this article, the technical roadmap is shown in Figure 2 below. We trained three networks, lraspp, U-Net, and ECA-UNet, with existing public datasets. The performance of the three networks was analyzed based on the data obtained from the network training, and the effects before and after adding attention mechanisms were compared. Finally, actual cracks were used for image-segmentation and recognition–visualization comparison analysis in the real structure.

## 3. Evaluation of Attention-Modified DNN

In this section, we conducted training and testing on three different deep-learning network models under the same conditions of batch size, learning rate, and iteration number. The study aimed to investigate the testing performance of different models based on their training results.

### 3.1. Evaluation Metrics

To verify the training effect of different models, we used precision, recall, and intersection over union (IOU) as evaluation metrics. Recall is the proportion of true positive samples in the model-predicted positive samples, usually indicating the model’s recall performance, as shown in Formula (2); precision represents the proportion of true positive samples predicted by the model to be positive, as shown in Formula (3); IOU represents the degree of overlap between different class samples and labels, as shown in Formula (4).
recall = TP/(TP + FP)(2)
precision = FP/(TP + FP)(3)
IOU = TP/(TP + FN + FP)(4)

In the formulas, TP denotes the number of true positive samples that were predicted correctly, and FP is the number of false positive samples that were predicted incorrectly. TN is the number of true negative samples that predicted correctly, and FN is the number of false negative samples that were predicted incorrectly.

### 3.2. Training and Analysis of the Original Model

This section first compares the performance of the U-Net network and the lraspp network. The training set used 5000 images with and without cracks from the bridge-crack library [47], with a crack-to-non-crack image ratio of 4:1. A validation set of 1000 crack images was used. The training dataset for the crack images included cracks in vertical, horizontal, and diagonal orientations. Part of the dataset is shown in Figure 3. Each crack image had a size of 256 × 256 pixels. After fine annotation with annotation tools, each crack image was paired with a PNG data label corresponding to the JPG format original image. All three models were iterated 200 times, and the training results are shown in Table 1. The lraspp model did not perform better overall than U-Net on the test set. In terms of precision, the value for lraspp was 0.810 and for U-Net was 0.921, indicating that lraspp was 0.111 less precise than U-Net. In terms of IOU, lraspp was 0.097 less than U-Net, and in terms of recall, lraspp was 0.051 less than U-Net, with lraspp having a value of 0.588 and U-Net having a value of 0.639.

Based on the above data, U-Net performs better than lraspp in all three indicators. Therefore, we use the U-Net network with attention mechanism in the following text.

### 3.3. Improved U-Net Model Based on Attention Mechanism

The ECA-UNet after adding the attention mechanism is shown in Figure 4. On the original U-Net structure, we added the attention mechanism ECA to the sampling part of each layer. Because the downsampling part is the main feature-extraction network, adding the attention mechanism to the trunk-extraction part of the downsampling part will interfere with the weight of the original network on image–feature extraction; furthermore, it will cause the model to be unable to accurately judge and distribute the features. Therefore, we added the ECA attention mechanism to enhance the feature pick-up network; that is, by up-sampling each layer, a total of 4 points was added. The feature graph optimized by the attention mechanism was then fused with the five effective feature layers obtained by the backbone network, and finally the classification output was obtained through 1 × 1 convolution.

The training results are displayed in Figure 5, Figure 6 and Figure 7 below, where Figure 5, Figure 6 and Figure 7 show the precision, recall, and IOU curves of the three models after training. From the precision curve, it can be observed that lraspp oscillates around 65% without showing a significant upward trend. While ECA-UNet and U-Net overlap in the early stages, ECA-UNet has a tendency to oscillate downwards compared to U-Net after about 140 epochs, although both maintain around 80%. From the recall curve, it can be seen that lraspp performed well in terms of recall, but with more spikes in the curve than the other two models. A clear restrict relationship between recall and precision is noticeable, meaning that the model did not balance the relationship between these two indicators well. However, ECA-UNet performed better than U-Net after about 150 epochs, showing a gradual upward trend. From the IOU curve, it can be seen that lraspp performed poorly, not as well as the other two networks. The ECA-UNet curve overlaps with the U-Net curve to a high degree, and the upward trend is similar. Based on the above analysis, lraspp performed poorly in both metrics and did not balance the relative relationship between precision and recall well. On the other hand, ECA-UNet performed well in terms of recall and IOU, especially surpassing U-Net in terms of recall.

The improved ECA-UNet was trained with the same parameters as the above model, and the performance comparison with U-Net is shown in Table 2 as follows:

According to the table, the ECA-UNet scores 0.692 in the IOU metric, which is 0.016 higher than U-Net, and scores 0.770 in the recall metric, which is 0.131 higher than U-Net. The improved recall rate has been increased, improving the comprehensiveness of crack identification. The improved model shows a decrease in precision compared to the original model. This is because recall and precision have a constraining relationship, where an increase in one can lead to a decrease in the other. Therefore, we needed to balance these two indicators under existing conditions and ensure an improvement in the indicators while maintaining a relatively balanced state. Due to the large and diverse training dataset, as well as the unfamiliarity of the model with the features required by our needs during the process of learning-feature extraction, the model tended to have a higher error rate in recognition. On the other hand, the testing dataset had fewer images, and the model had already completed the learning process, gaining a better understanding of the desired features. Therefore, there could be cases where the metrics in the testing results are higher than the training output metrics. However, the training and testing data do not overlap, and the numerical relationship between the two does not have a significant correlation. When evaluating the performance of the model, it is insufficient to compare the testing results alone, as the training data do not affect the assessment of the model’s recognition effectiveness. In terms of model running speed, under the same conditions, the testing time for a single image in both U-Net and ECA-U-Net is 0.058 s, while the testing time for lrassp is 0.062 s. This further highlights the advantages of ECA-U-Net, which has high accuracy and fast running speed.

## 4. Field Test of Raw Structural Crack Images

To more intuitively demonstrate the performance of each model in recognizing cracks, we used real-life structural cracks for crack recognition, which contained noise motifs other than cracks. The crack in Figure 8a is slightly inclined, with a physical width of 1.5 mm. The crack in Figure 8b is almost horizontal, with a physical width of 2.1 mm, and the crack in Figure 8c is vertical, with a physical width of 1.7 mm. Furthermore, the three cracks have disconnected points along the paths themselves, respectively. The raw structural images contain multiple noise motifs including water stains, spots, joint lines, concrete stripes, scratches, pits, etc., as marked with blue boxes. These images came from an on-site bridge structural inspection and were not contained in the bridge-crack library [47]. The results of recognition are shown in Figure 8.

It can be seen that although the lraspp network was not cheated by all kinds of noise motifs in the raw images, it recognized very few parts of the crack regions. It missed the majority of the crack regions in Figure 8b,c, and it missed all the crack regions in Figure 8a. As for the U-Net, it almost successfully detected all the crack regions in all three of the testing images. Nonetheless, U-Net’s recognition results contained a small number of non-crack noise motifs. Part of the joint line in Figure 8a was misidentified as cracks, and the concrete strips and the pitted area in Figure 8b were also misidentified as cracks. Moreover, part of the water stains and the concrete strip in Figure 8c were mistakenly recognized as cracks. This indicated that U-Net could achieve satisfactory performance in detecting the cracks but the robustness against noise motifs was not enough. When it came to the ECA-UNet, which was based on the U-Net and modified by adding the attention mechanism, a better performance was achieved. It accurately recognized the crack areas in the raw crack images. Yet, part of the concrete strip in Figure 8c still caused error detection.

Seen from the result, the recognition performance of lraspp on the feature of cracks is unsatisfactory, and the ability to capture features is weak. The U-Net network can successfully identify cracks, but it is still affected by some false crack noise interference with crack-like characteristics, such as the concrete strips and some of the water stains. The improved ECA-UNet based on the attention mechanism can accurately identify the crack area and is more robust against all kinds of noise motifs, although errors can still be detected.

## 5. Conclusions

This article investigated the performance of the ECA attention mechanism in improving the crack-detection capacity of the deep neural network. Three trained models were evaluated for their recognition performance on crack images of actual bridges. The following conclusions were drawn:(i)Training of existing network public datasets: This article discusses training we conducted on two primary models, lraspp and U-Net, using a publicly available dataset of bridge cracks. The trained models were then tested for their generalization performance, and the results showed that the U-Net model performed better than the lraspp model in terms of data metrics. The precision, recall, and IOU values of the U-Net model were 0.111, 0.051, and 0.097 higher than those of the lraspp model, respectively.(ii)Improvement of the U-Net network based on the ECA attention mechanism: U-Net performed well on the crack dataset, and based on this, an ECA attention mechanism was added to the upsampling part of the U-Net network to enhance the model’s crack-detection performance. By keeping the original training parameters unchanged, the results of the training showed an increase of 0.131 in the recall rate and an improvement of 0.016 in the IOU compared to the original U-Net network, achieving improvements in both performance metrics.(iii)Recognition of real structural cracks in raw images: In the recognition of actual structural crack images, it was observed that the lraspp network was almost insensitive to the crack feature and recognized hardly any cracks. Although the U-Net network was able to identify cracks, it also misjudged some false crack noise. The improved ECA-UNet network proposed in this paper showed better recognition performance than the other two networks and accurately identified cracks without minor mistakes.(iv)This paper proposed a method to improve the crack-detection performance of the original U-Net model by integrating the ECA attention mechanism. Although the ECA-UNet achieved comparatively satisfactory results, more efforts are still required to improve the crack-detection performance. As for the improvement of detection performance, it can be seen from the testing results that the ECA-UNet can be cheated by noise motifs with linear geometry. Thus, the proposed network needs to be trained for robustness to exclude images with crack-like linear noise motifs. Furthermore, the network is quite large in terms of training parameters; therefore, how to reduce the size of the network and keep the crack detection performance is also an important aspect waiting for investigation. Moreover, attempts to embed the existing models into mobile devices for real-time crack identification are also pertinent to bringing this method into practical application.

## Figures and Tables

**Figure 1 sensors-23-06295-f001:**
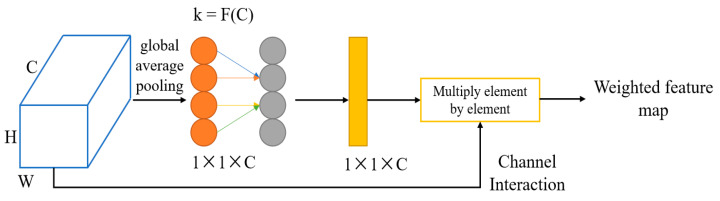
Diagram of the ECA structure.

**Figure 2 sensors-23-06295-f002:**
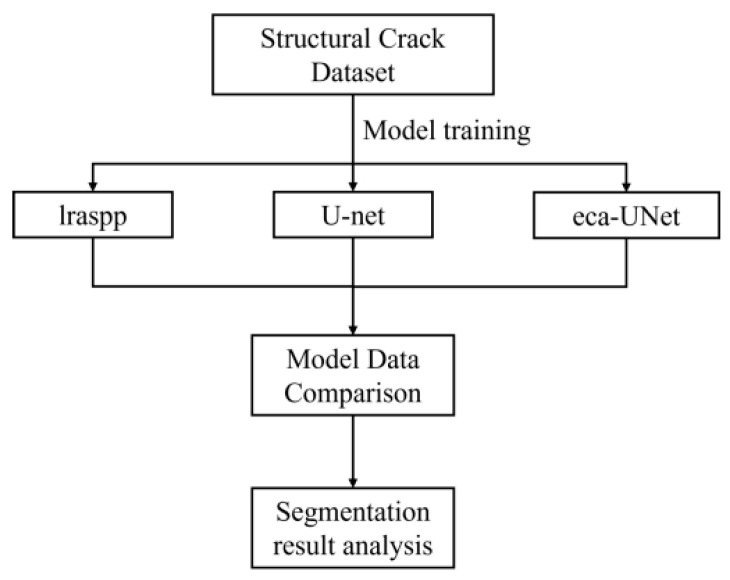
Technical roadmap.

**Figure 3 sensors-23-06295-f003:**
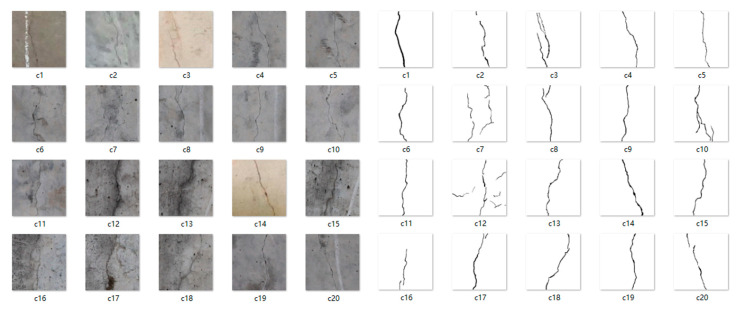
Example of the crack dataset.

**Figure 4 sensors-23-06295-f004:**
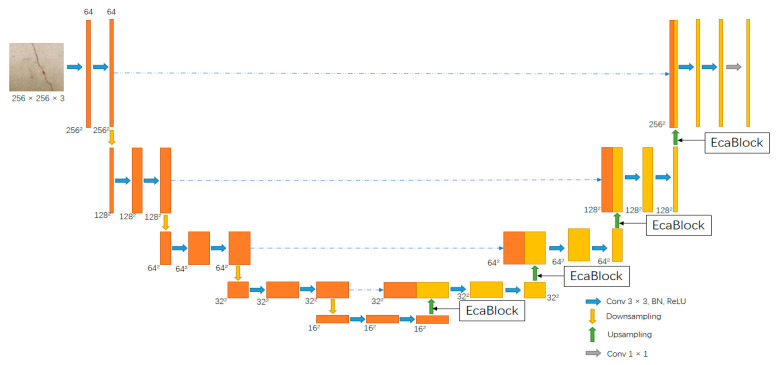
The schematic diagram of the improved ECA-UNet structure is shown as above.

**Figure 5 sensors-23-06295-f005:**
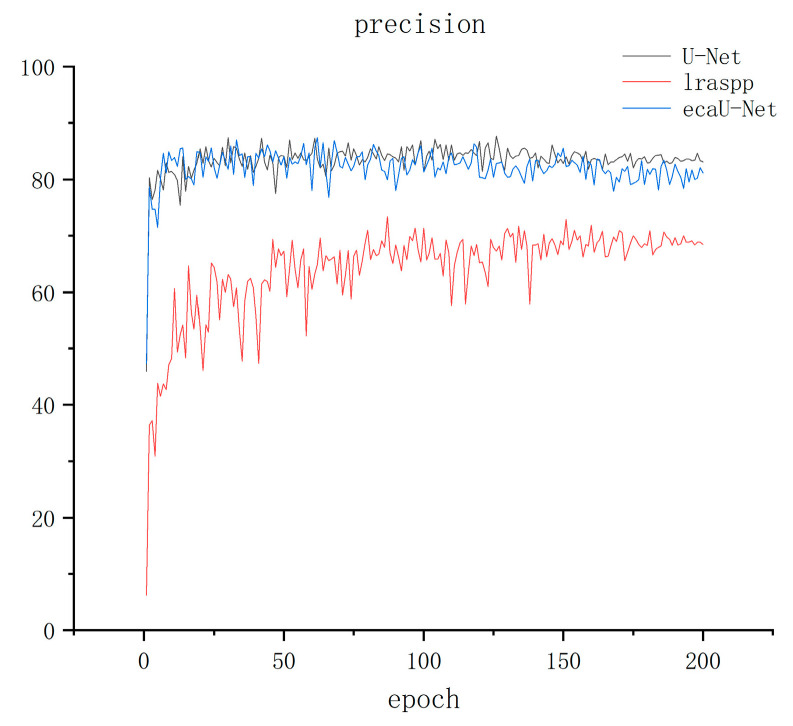
Precision metric result graph.

**Figure 6 sensors-23-06295-f006:**
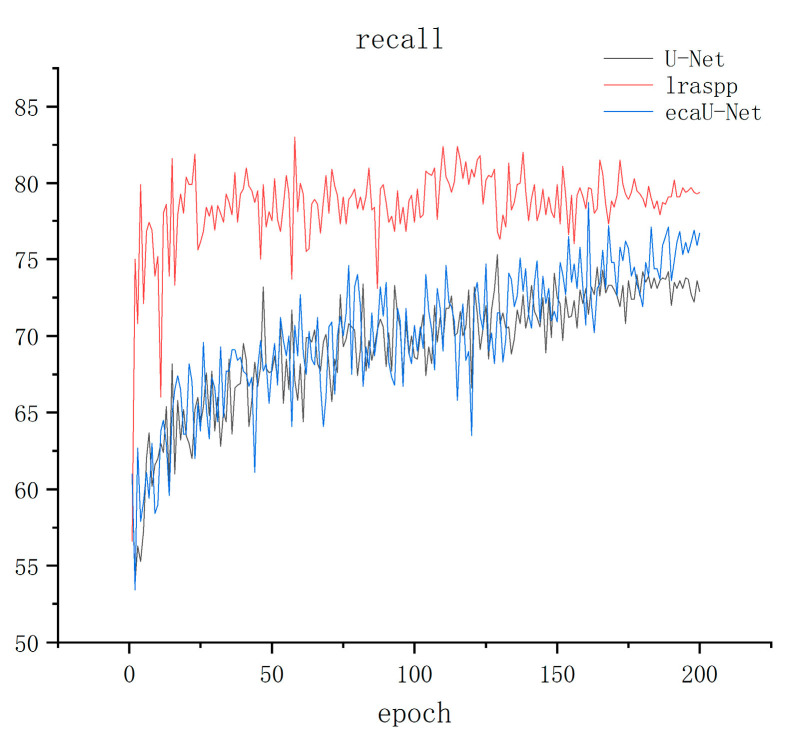
Recall metric result graph.

**Figure 7 sensors-23-06295-f007:**
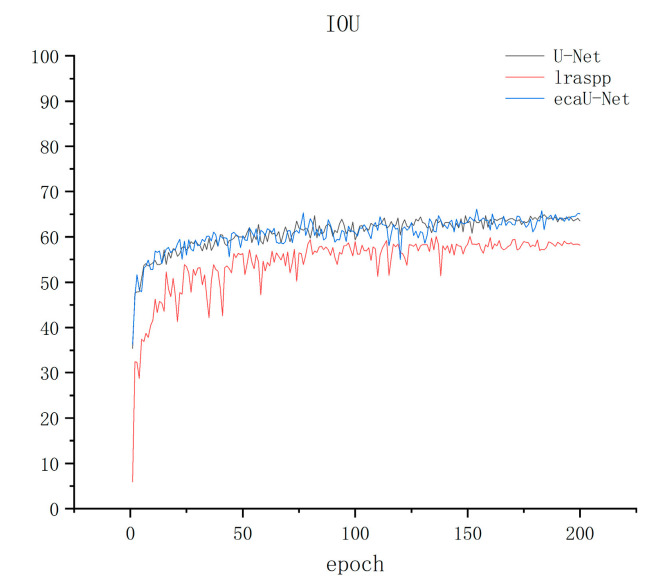
IOU metric result graph.

**Figure 8 sensors-23-06295-f008:**
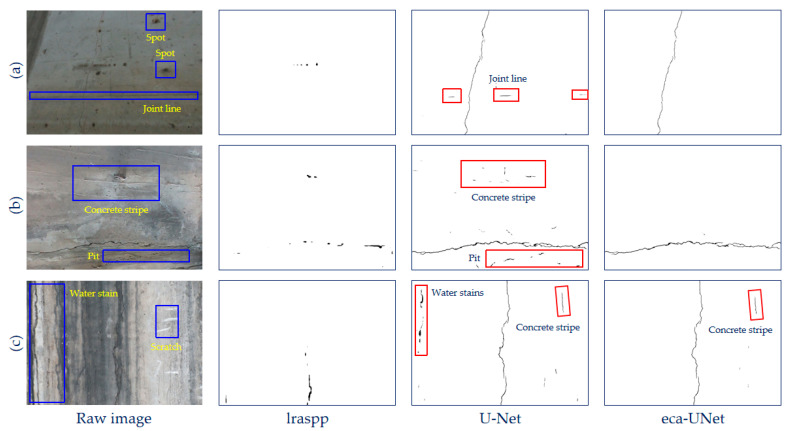
Recognition results of actual crack images by the model.

**Table 1 sensors-23-06295-t001:** Model part of the training data.

Model Name	Precision	IOU	Recall
U-Net	0.921	0.676	0.639
lraspp	0.810	0.579	0.588

**Table 2 sensors-23-06295-t002:** U-Net and ECA-UNet result comparison.

Model Name	Precision	IOU	Recall
U-Net	0.921	0.676	0.639
ecaUNet	0.872	0.692	0.770

## Data Availability

The data that support the findings of this study are available from the corresponding author upon reasonable request.

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
