# Peer review of "Modification and Evaluation of Attention-Based Deep Neural Network for Structural Crack Detection"

_sensors, 2023, doi:10.3390/s23146295_

Round 1
Reviewer 1 Report
The authors have taken a binary classification of crack identification considering image data. The authors have compared lraspp and U-Net, using a publicly available dataset of bridge cracks and concluded U-Net is better. Further, they integrated U-Net with ECA attention mechanism for more accuracy. The following queries need to be addressed
1. In order to highlight the advantage, the author may compare computational time between i) U-Net ii) lraspp iii) U-Net with ECA.
2. The author should consider imbalance dataset in training of two classes and compare the three techniques and plot the results in terms of confusion matrix, recall, precision etc.
3. The authors may present in detail on ECA attention mechanism and the necessity to augment with U-Net, why author attention mechanisms (like encoder-decoder ) are not considered.
4. Pls. explain mathemically and also with pyhsical insights how ECA attention mechanism makes U-Net intrepretable
5. The contribution, novelty can be clearly mentioned in the last paragraph of the introduction. The organization of the paper can also be mentioned.
It can be improved.
Reviewer 2 Report
Check my comments in the attached PDF file

Minor check of english is requred
Round 2
Reviewer 1 Report
Accept
Author Response
Thanks for the comment.
Reviewer 2 Report
The revised version of the paper has been improved. Hence, the paper can be accepted for publication.
Author Response
Thanks for the comment.
